# Deep Coupling Recurrent Auto-Encoder with Multi-Modal EEG and EOG for Vigilance Estimation

**DOI:** 10.3390/e23101316

**Published:** 2021-10-09

**Authors:** Kuiyong Song, Lianke Zhou, Hongbin Wang

**Affiliations:** 1College of Computer Science and Technology, Harbin Engineering University, Harbin 150001, China; songkuiyong@hrbeu.edu.cn (K.S.); zhoulianke@hrbeu.edu.cn (L.Z.); 2Department of Information Engineering, Hulunbuir Vocational Technical College, Hulunbuir 021000, China

**Keywords:** vigilance estimation, electroencephalogram, electrooculogram, deep coupling recurrent auto-encoder, multi-modal fusion

## Abstract

Vigilance estimation of drivers is a hot research field of current traffic safety. Wearable devices can monitor information regarding the driver’s state in real time, which is then analyzed by a data analysis model to provide an estimation of vigilance. The accuracy of the data analysis model directly affects the effect of vigilance estimation. In this paper, we propose a deep coupling recurrent auto-encoder (DCRA) that combines electroencephalography (EEG) and electrooculography (EOG). This model uses a coupling layer to connect two single-modal auto-encoders to construct a joint objective loss function optimization model, which consists of single-modal loss and multi-modal loss. The single-modal loss is measured by Euclidean distance, and the multi-modal loss is measured by a Mahalanobis distance of metric learning, which can effectively reflect the distance between different modal data so that the distance between different modes can be described more accurately in the new feature space based on the metric matrix. In order to ensure gradient stability in the long sequence learning process, a multi-layer gated recurrent unit (GRU) auto-encoder model was adopted. The DCRA integrates data feature extraction and feature fusion. Relevant comparative experiments show that the DCRA is better than the single-modal method and the latest multi-modal fusion. The DCRA has a lower root mean square error (RMSE) and a higher Pearson correlation coefficient (PCC).

## 1. Introduction

The fatality rate of traffic accidents is very high. According to statistics, millions of people die from traffic accidents every year. Tired driving and inattention are the main causes of traffic accidents. Modern sensor technology has been widely used in driver condition monitoring, and has reduced traffic accidents to a certain extent and saved thousands of lives [1].

A portable wearable device can collect electroencephalography (EEG) and electrooculography (EOG) signals, which are used to evaluate the driver’s state in real time [2,3,4]. EEG signals can directly reflect the activity of the human brain and capture the changes of radio waves caused by fatigue or drowsiness [5]. EEG is a promising neurophysiological indicator that has distinguished wakefulness from sleeping in various studies. EOG measures the potential between the front and back positions of human eyes, which contains information about vigilance and eye movement, the latter of which is an effective indicator of human psychological activities. EEG and EOG signals come from different sensors, and such data are called multi-modal data. Multi-modal fusion methods include the multi-core method, the graph model, and the neural network method [6]. In recent years, multi-modal data fusion has attracted extensive attention [6,7,8], and the fusion of vibration signals and acoustic signals with different attributes and characteristics provides better fault diagnosis results [9,10]. In the field of artificial intelligence, the fusion of image, sound, text, and video is a current research hotspot [11,12,13].

EEG and EOG represent internal cognitive state and external subconscious behaviors, respectively, and the information gathered by these two modes is closely connected and complementary. Related studies have shown that the fusion of EEG and EOG in alertness analyses has obvious advantages over the use of each of them alone [4,11,12,13,14]. However, there are some difficulties regarding the integration of EEG and EOG. On the one hand, there are some human disturbances within the data itself. In the process of data monitoring, the unconventional actions and thinking changes of the tested body make data noise, which is difficult to uncover. On the other hand, multi-modal analysis of biological signals is very difficult, and it is a challenging task to identify complementary and contradictory information from the available signals. In addition, the lack of an ideal synchronization method between modes is another challenge related to multi-modal fusion analysis.

A number of machine learning methods based on EEG and EOG have been proposed for vigilance estimation. For example, support vector regression (SVR) was applied to EEG, EOG, and multi-modal EEG and EOG, and was used as a benchmark to evaluate other models [2]. Vigilance is a dynamic process because the user’s internal psychological state is involved in time evolution. In order to incorporate time dependence into vigilance estimation, continuous conditional neural field (CCNF) and continuous conditional random field (CCRF) were introduced in [2] to construct a vigilance estimation model. The authors of [3] proposed a multi-modal fusion strategy that uses the depth auto-encoder model to learn better sharing. The authors of [4] put forward a method of an adversarial domain adaptive network for reusing data, which saves the time of labeling a large amount of data. Huo [15] used the discriminative graph regularized extreme learning machine (GELM) to evaluate the driver’s state. An extreme learning machine is an efficient and practical feedforward neural network with a single hidden layer. The authors of [16] put forward a continuous vigilance estimation method using long- and short-term memory (LSTM) neural networks and combining EEG and EEG signals from the forehead. This method explores time-dependent information and significantly improves the performance of vigilance estimation. The authors of [17] proposed a double-layered neural network with subnetwork nodes (DNNSN), which is composed of several subnet nodes, and each node is composed of many hidden nodes with various feature selection capabilities. Zhang [14] suggested that the capsule attention model and deep LSTM should be integrated with EEG and EEG. The capsule attention model learns the temporal and hierarchical/spatial dependencies in the data through the LSTM network and the capsule feature representation layer.

In recent years, deep neural networks have been widely studied regarding the fusion of EEG and EGG, and promising results have been achieved [13]. The convolutional neural network (CNN), recurrent neural network (RNN), auto-encoder, anti-neural network, and attention model are widely used in feature extraction and the fusion of EEG and EOG. In terms of image reconstruction and image fusion, auto-encoders and convolutional neural networks also show their advantages. The authors of [18] showed that even simple autoencoders can be trained to reconstruct an image in such a way that the human eye would not be able to distinguish the noise and signal from a damaged sample. Professor Lu Baoliang of Shanghai Jiao Tong University and his team have done a lot of work regarding the integration of EEG and EOG. They have done relevant simulation tests, collected a large amount of test data, and put forward a series of vigilance evaluation methods [2,4,19,20,21] based on these test data. The experimental data used in this study came from Lu Baoliang’s team.

Auto-encoders can extract deep features of data and remove noise interference. An RNN with a memory function has a good effect on processing time-series data, and it does not require the high synchronization of time. We designed a feature extraction and fusion framework—a deep coupling recurrent auto-encoder (DCRA) model, which can effectively solve the above problems. Our contributions include the following:The DCRA uses multi-layer gated recurrent units (GRUs) to extract deep features and uses the joint objective loss function to fuse them together.The joint loss function uses Euclidean distance similarity metrics in a single mode, and the multi-modal loss is measured by a Mahalanobis distance of metric learning [22,23], which can effectively reflect the distance between different modal data so that the distance between different modes can be described more accurately in the new feature space based on the metric matrix, and the losses of the two modes are summed according to weights.Compared to the latest fusion method and the single-modal method, the method proposed in this paper has a lower root mean square error (RMSE) and a higher Pearson correlation coefficient (PCC).

The remainder of this paper is organized as follows: In Section 2, the auto-encoder and metric learning are described. The deep recurrent auto-encoder is then extended to a deep coupling recurrent auto-encoder and a combinational model. The experimental data and evaluation methods are introduced in Section 3. Section 4 describes the experimental results and compares the performance of different models. Conclusions are presented in Section 5.

## 2. Materials and Methods 

### 2.1. Auto-Encoder

An auto-encoder is an unsupervised symmetric structure neural network model [24]. It consists of two parts—an encoder that converts inputs into potential representations, and a decoder that converts internal representations into outputs. An auto-encoder usually has the same number of neurons in the input and output layers. If the number of neurons in the hidden layer is smaller than that in the input and output layers, the hidden layer needs to learn the most important features of the input data and delete the unimportant features. The output layer is often called reconstruction, and the auto-encoder attempts to reconstruct the input through the loss function so that the input and output are as similar as possible. The auto-encoder encoding function *f* and decoding function *g* are as follows:(1)$$y=f(x)=s_{1}(w^{\left(1\right)}x+b^{\left(1\right)}),$$
(2)$$z=g(y)=s_{2}(w^{\left(2\right)}f(x)+b^{\left(2\right)}),$$
where *w*^(1)^ and *w*^(2)^ are weight matrices, *b*^(1)^ and *b*^(2)^ are bias vectors, and *S*_1_ and *S*_2_ are nonlinear activation functions. The objective of an auto-encoder is to optimize *w* and *b* to minimize the reconstruction error. Traditionally, the mean squared error or cross entropy is used to compute the reconstruction error. The mean squared error function is *l_MSE_*(*x*,*z*) = $\|x-z{\|}_{2}^{2}$, and the cross entropy function is
(3)$$l_{CE}(x,z)=-\sum_{k=1}^{d}x_{k}\log z_{k}+(1-x_{k})\log(1-z_{k})$$

The auto-encoder minimizes the objective function by optimizing *w* and *b* to maximize the similarity of the reconstructed data input. 

### 2.2. Metric Learning

Mahalanobis distance was proposed by Indian statisticians in the literature [22] to represent the covariance distance of data. Compared with Euclidean distance, Mahalanobis distance has some excellent properties of its own—it is independent of decoupling and the dimension. The traditional Mahalanobis distance based on the inverse of the covariance matrix is usually used to reflect the internal aggregation relationship of data. However, in many classification tasks, it is not enough to measure the distance function only to reflect the internal aggregation relationship of data, and it is more important to establish the relationship between sample attributes and categories. In the study of metric learning, Mahalanobis distance is no longer simply limited to the inverse of the covariance matrix, but needs to be obtained through the process of metric learning [25,26,27]. If there are two sequences *x_i_* and *x_j_*, then a semi-positive definite matrix *M* is given, called the Mahalanobis distance, expressed as follows:(4)$$D_{M}(x_{i},x_{j})={\left(x_{i}-x_{j}\right)}^{T}M(x_{i}-x_{j})$$

Distance metric learning refers to using a given training sample set to learn a metric matrix that can effectively reflect the distance between data so that in the new feature space based on the metric matrix, the distribution between similar samples is more compact, while the distribution between different samples is more distant. Metric learning is to learn *M*. In order to ensure a non-negative distance and satisfy the triangle inequality, *M* must be a (semi) positive definite symmetric matrix, that is, there is an orthogonal basis *P* so that *M* can be written as *M* = *PP^T^*. Commonly used distance measurement learning methods include the probabilistic global distance metric (PGDM) [26], the large margin nearest neighbor (LMNN) [28], and information-theoretic metric learning (ITML) [29]. In this study, Mahalanobis distance was used to measure the similarity between multiple modes.

### 2.3. Deep Coupling Recurrent Auto-Encoder (DCRA)

The deep recurrent auto-encoder extracts deep features, which is the basis of feature fusion. The coupling layer connects two single auto-encoders together. The DCRA integrates feature representation and feature fusion.

#### 2.3.1. Coupling Auto-Encoder

Multi-modal EEG and EEG information is closely related and complementary. It is one of the strategies of multi-modal fusion to strengthen the common features of multi-modal data and weaken the individual features of multi-modal data. The structure of the coupling auto-encoder is shown in Figure 1. The coupling auto-encoder consists of two auto-encoders of the same structure. The inputs of the two auto-encoders are EEG and EOG data, and the model reconstruction is consistent with the inputs. The coupling layer fuses two single-modal features together through a joint objective loss function [11]. Although the two auto-encoders have the same structure, the parameters of the two auto-encoders are different because of different inputs.

In this study, a joint objective loss function was designed to train the coupling auto-encoder [10,11]. The joint loss function is shown in Formula (5). The joint loss function is composed of three parts: EEG loss *L_E_,* EOG loss *L_O_*, and multi-modal loss *S*. *L_E_* and *L_O_* are single-modal losses. Considering the internal correlation of a single mode, a single mode uses Euclidean distance to measure similarity, as shown in Formulas (6) and (7), *z_E_* and *z_O_* are the reconstruction data from the inputs *x_E_* and *x_O_*. The use of Euclidean distance for multi-modal loss *S* between two different modes cannot fully reflect the internal relationship between them. In order to learn the internal relations between the two modes, Mahalanobis distance was used to measure different modes, and this distance was obtained through metric learning. The Mahalanobis distance of metric learning can reflect the internal relations between different modes and express the differences between them [26,27]. As shown in Formula (8), *M* is the Mahalanobis distance of metric learning.
(5)$$L(x_{E},x_{O};\theta_{E},\theta_{O})=(1-\alpha )(L_{E}(x_{E},\theta_{E})+L_{O}(x_{O},\theta_{O}))+\alpha S(x_{E},x_{O};\theta_{E},\theta_{O})$$
(6)$$L_{E}(x_{E},\theta_{E})=∥x_{E}-z_{E}){∥}^{2}$$
(7)$$L_{O}(x_{O},\theta_{O})=∥x_{O}-z_{O}){∥}^{2}$$
(8)$$S(x_{E},x_{O};\theta_{E},\theta_{O})=f_{E}(x_{E},\theta_{E})Mf_{O}(x_{O},\theta_{O})$$

In Formula (5), *L_E_*(*x_E_*,*θ_E_*) and *L_O_*(*x_O_*,*θ_O_*) are auto-encoder loss functions of EEG and EOG, respectively; *θ_E_* and *θ_O_* represent parameters of corresponding models. α is the weight of multi-modal loss in the joint loss function. If α = 0, the joint loss function degrades to the loss function of a single auto-encoder, which cannot capture any correlation between inputs from different modes. If α = 1, any pair of multi-modal inputs has a correlation. In short, the loss function only focuses on the constraint of correlation and completely ignores the characteristics of the data.

In Formula (8), *f_E_*(*x_E_*,*θ_E_*) and *f_O_*(*x_O_*,*θ_O_*) represent the auto-encoder mapping functions of EEG and EOG, and *M* in Formula (8) is the Mahalanobis distance for metric learning. In this study, the probabilistic global distance metric learning (PGDM) method was used to learn the Mahalanobis matrix [26]. This method transforms the metric learning method into a convex optimization problem with constraints, and takes the selected pair constraint as the constraint condition of training samples. The dominant idea was to minimize the distance of sample pairs of the same category while constraining the distance between different samples of the same category that were greater than a certain value. The optimization model is as follows:(9)$$\begin{array}{l} \underset{M}{min}\sum_{(x_{i},x_{j})\in S}||x_{i}-x_{j}|{|}_{M}^{2}\\ s.t.\sum_{(x_{i},x_{j})\in D}||x_{i}-x_{j}|{|}_{M}\geq 1,M\underline{≻}0 \end{array}$$

If *M* is the Mahalanobis distance to be learned, to minimize the sum of the squares of *x_i_* and *x_j_* distances for any homogeneous *x_i_* and *x_j_*, the limiting condition is that the distance between *x_i_* and *x_j_* of different classes is greater than 1 and *M* is semi-positive. In Formula (9), *S* is the same classes of data, and *D* is the different classes of data. The loss function of PGDM is expressed as the following:(10)$$g(M)=g(M_{11},\cdots ,M_{nn})=\sum_{(x_{i},x_{j})\in S}||x_{i}-x_{j}|{|}_{M}^{2}-\log\left(\sum_{(x_{i},x_{j})\in D}||x_{i}-x_{j}|{|}_{M}\right)$$

This loss function is equivalent to the optimization model, which is a convex optimization problem and can be directly optimized by Newton and quasi-Newton methods. Compared with Euclidean distance, Mahalanobis distance can describe the relationship between two different modes more accurately.

#### 2.3.2. Deep Coupling Recurrent Auto-Encoder (DCRA)

A deep coupling recurrent auto-encoder can extract deep features, which are the basis of feature fusion. In this paper, a deep coupling recurrent auto-encoder (DCRA) is proposed. Its structure is shown in Figure 2. DCRA encoding and decoding are composed of three layers of gated recurrent units (GRUs), and the coupling layer connects two independent auto-encoders. The DCRA is trained by the joint objective loss function, as expressed by Formula (5).

A recurrent neural network (RNN) is a neural network with short-term memory and parameter sharing. The nodes between the hidden layers in the structure of a recurrent neural network are connected, and the input of the hidden layer includes not only the output of the input layer but also the output of the hidden layer at the last moment. This structure focuses on the relevance of data before and after, and is particularly suitable for video, voice, text, and other time-series-related problems. As shown in Figure 3, the recurrent neural network is connected not only between adjacent layers but also between hidden layers.

In Figure 3, at each time step *t*, neuron *y*(*t*) receives input vector *x*(*t*) and output vector *y*(*t* − 1) of the previous time step, and transmits backwards, step-by-step, as expressed by Formula (11):(11)$$y_{\left(t\right)}=\phi (W_{x}^{T}x_{\left(t\right)}+W_{y}^{T}y_{\left(t-1\right)}+b)$$

In Formula (11), *W_x_* and *W_y_* are the weight matrices of input *x*(*t*) and *y*(*t* − 1), *b* is the bias vector, and *ϕ* is the activation function. The parameters of the recurrent neural network are learned by the back propagation algorithm, that is, the errors are passed forward, step-by-step, in the reverse order of time. Because the data transforms when traversing the RNN, some information is lost at each time step, resulting in a worse final result. Long- and short-term memory (LSTM) has been proposed to solve this problem [30,31,32]. The unique structure of LSTM can detect medium and long-term dependence of data. A GRU is a simplified version of an LSTM recurrent neural network [33]. The structure of a GRU is shown in Figure 4. We used a GRU in this study to design the recurrent auto-encoder.

In Figure 4, *z*(*t*) controls the forgetting gate and the input gate. Some parts of the forgetting gate that control the long-term state should be deleted. The input gate control should add some parts of *g*(*t*) to the long state. If the gate controller outputs 1, the forgetting gate opens and the input gate closes. If 0 is the output, the opposite is true. Gate controller *r*(*t*) controls the display of a portion of the previous state to the main output layer *g*(*t*). *g*(*t*) is the main output layer, whose function is to analyze the current input *x*(*t*) and the previous state *h*(*t* − 1), store the most important part in the long-term state, and its output goes directly to *h*(*t*).
(12)$$z_{\left(t\right)}=\sigma (W_{xz}^{T}x_{\left(t\right)}+W_{hz}^{T}h_{\left(t-1\right)}+b_{z})$$
(13)$$r_{\left(t\right)}=\sigma (W_{xr}^{T}x_{\left(t\right)}+W_{hr}^{T}h_{\left(t-1\right)}+b_{r})$$
(14)$$g_{\left(t\right)}=\tanh(W_{xg}^{T}x_{\left(t\right)}+W_{hg}^{T}(r_{\left(t\right)}⊗h_{\left(t-1\right)}+b_{g})$$
(15)$$h_{\left(t\right)}=z_{\left(t\right)}⊗h_{\left(t-1\right)}+(1-z_{\left(t\right)})⊗g_{\left(t\right)}$$

In Formulas (12)–(14), *W_xz_*, *W_xr_*, and *W_xg_* are the weight matrices of each layer in 3 layers connected with input vector *x*(*t*). *W_hz_*, *W_hr_*, and *W_hg_* are the weight matrices of the connection between each layer and the previous short-term state *h*(*t* − 1) in the three layers. *b_z_*, *b_r_*, and *b_g_* are the offset terms of each of the three layers. GRUs can learn to identify important inputs, store them in a long-term state, and extract them when needed, which is the advantage of a GRU processing time-series data.

DCRA model training is stratified from bottom to top. The DCRA is able to learn similar features from different modes by mapping multi-mode signals into the same representation space. The DCRA algorithm is shown below (Algorithm 1).


**Algorithm 1: DCRA.**
      Input: training data EEG, EOG, Mahalanobis distance *M*
      Output: DCRA model parameters
1    Training layer 1 GRU;
2    Training layer 2 GRU;3    Training layer 3 GRU;4    Update DCRA parameters using Formula (5) using gradient descent;5    Repeat steps 1–4 until the model converges.

## 3. Data Description and Evaluation Measures

### 3.1. Dataset

SEED-VIG [2] is a vigilance evaluation dataset collected in a simulation experiment by Lu Baoliang’s research group. In the experiment, a neuroscan system and eye-tracking glasses were used to record the real-time data of the tester. The neuroscan system recorded EEG and EOG data, and the eye-tracking glasses recorded eye movement data, which included saccade, fixation, blinking, and closing eyes. Testers needed to drive the simulated car in a virtual environment for 120 min without any warning or interference during driving. Testers were required to test after lunch because drivers are prone to drowsiness at this time. SEED-VIG recorded data from 23 testers. The testers included 12 women and 11 men. The eye-tracking glasses accurately captured information about their eye movements. Eye movement-based PERCLOS [34] is considered to be the most reliable and effective method for measuring driver alertness levels. PERCLOS is equal to (blink + eyes CLOSures(CLOS))/(blink + fixation + saccade + eyes CLOSures(CLOS)). This is a widely accepted indicator of alertness.

### 3.2. Evaluation Methods

PERCLOS is suitable for regression analysis, and the root mean square error (*RMSE*) is a common evaluation method for regression models [35]. The *RMSE* uses the square error of the real value y and the predicted value $\hat{y}$ to evaluate the model, and the *RMSE* formula is as follows:(16)$$RMSE(y,\hat{y})=(\frac{1}{n}\sum_{i=1}^{n}(y_{i}-{\hat{y}}_{i}{)}^{2}{)}^{1/2}$$

The *RMSE* does not provide some structural information, while the Pearson correlation coefficient (*PCC*) provides an assessment of the linear relationship between the predicted and true values. The *PCC* is used as a complement to the RMSE in related models for assessing EEG and EOG. The formula of *PCC* is as follows:(17)$$PCC(y,\hat{y})=\frac{\sum_{i=1}^{n}\left(y_{i}-\bar{y})({\hat{y}}_{i}-\hat{\bar{y}}\right)}{\sqrt{\sum_{i=1}^{n}{\left(y_{i}-\bar{y}\right)}^{2}\sum_{i=1}^{n}{\left({\hat{y}}_{i}-\hat{\bar{y}}\right)}^{2}}}$$$y={\left(y_{1},y_{2},\cdots ,y_{n}\right)}^{T}$ is the actual measured PERCLOS and $\hat{y}={\left({\hat{y}}_{1},{\hat{y}}_{2},\cdots ,{\hat{y}}_{n}\right)}^{T}$ is the model-predicted value. $\bar{y}$ and $\hat{\bar{y}}$ are the means of the real and predicted values, respectively. When the model is more accurate, the *PCC* value is larger and the *RMSE* value is smaller.

### 3.3. Comparison Method

In this study, eight feature level fusion methods were selected for comparison, including SVR, CCNF and CCRF [2], DAE [3], GELM [15], LSTM [16], DNNSN [17], and LSTM-CapsAtt [14]. The DNNSN proposes a double-layer neural network with subnetwork nodes. The model is composed of multiple subnet nodes, and each node is composed of many hidden nodes with various feature selection capabilities. The DNNSN demonstrated good results in an experiment. The LSTM-CapsAtt uses a capsule attention model and a deep LSTM network to fuse EEG and EOG. The capsule attention model uses LSTM and capsule feature representation layers to learn temporal and hierarchical/spatial dependencies in the data. Experiments have shown that the LSTM-CapsAtt achieves better results than the previous seven methods.

The DCRA was realized using the TensorFlow and Keras frameworks. In order to make the DCRA model more accurate, parameters, such as the number of layers of the DCRA model, the number of neurons in each layer, the activation function, the optimization function, and the learning rate, were debugged several times, and the most suitable parameter collocation was selected to achieve the best effect. The model parameter settings are shown in Table 1.

Before the experiment, the data was first standardized so that the data of different modalities were in the same range and equally distributed. In the experiment, we found that adding the batch normalization layer can improve the model performance, so the batch normalization layer was added after second, third, fifth and sixth layers of the model. The batch normalization layer realizes batch normalization, which has a positive effect on the deep network [36]. The learning rate and batch size were adjusted multiple times simultaneously, and the learning rate increased from 10^−5^ to 10, each time by 10 times. The batch size ranged from 16 to 256 and increased twofold each time. Finally, when the learning rate was 0.001 and the batch size was 32, the model performance was optimal and the convergence speed was the fastest. The activation function selected Relu and Sigmoid, and different layers selected different activation functions. Regarding the choice of optimizer, we tried AdaGrad, RMSProp, and Adam, and finally chose the more suitable Adam, which was faster than the other two optimizers.

### 3.4. Learning Mahalanobis Distance

In order to obtain the Mahalanobis distance *M*, some data was selected from the SEED-VIG dataset, and feature data was extracted with a single recurrent auto-encoder model to achieve data dimension reduction. Then, these characteristic data and labels were processed by the PGDM method for training and learning the Mahalanobis distance *M*. In this study, the PGDM algorithm was realized in MATLAB (Version 2016, MathWorks, Natick, MA, USA).

In reference [25], Mahalanobis distance and Euclidean distance were compared in UCR datasets, and the results show that Mahalanobis distance is smaller than Euclidean distance. These experiments show that Mahalanobis distance is more accurate than Euclidean distance in measuring the distance similarity of different modes.

## 4. Results and Discussion

### 4.1. Performance Analysis

In order to verify the effectiveness of the DCRA algorithm under different distance measures, DCRA_E, based on Euclidean distance, and DCRA_M, based on Mahalanobis distance, were compared with the other eight latest fusion methods. We employed five-fold cross-validation for the data, and no overlap existed between the testing and training data. As shown in Table 2, we used different algorithms to evaluate the values of RMSE and PCC on multi-modal EEG and EOG.

As can be seen in Table 2, the RMSEs of the time-dependent CCRF and CCNF methods were 0.10 and 0.095, respectively, and the PCCs 0.84 and 0.845, respectively. The DAE was almost the same as the CCNF without significant change. Although the LSTM cyclic neural network achieved an adequate PCC, the RMSE performance significantly reduced its actual performance. At the same time, we observed the fact that the combination of the ELM-based GELM model and the auto-encoder significantly improved the performance because it reduced the dimension of input data. The DNNSN performed better than LSTM, but not as well as the GELM in terms of the RMSE. The LSTM-CapsAtt made a great leap in performance, as the RMSE and PCC were 0.029 and 0.989, respectively, which is obviously better than the previous algorithms. Compared to the LSTM-CapsAtt, the DCRA_E was slightly insufficient, but the DCRA_M was better than the LSTM-CapsAtt, which shows that similarity measurements based on Mahalanobis distance have certain advantages.

In order to compare the performance of algorithms more accurately, the Friedman test was performed on some algorithms in Table 2. As shown in Table 3, the entire dataset was divided into five small datasets, namely *D*_1_, *D*_2_, *D*_3_, *D*_4_, and *D*_5_. We selected six algorithms with better results from the 10 algorithms in Table 2, performed five-fold cross-validation, and sorted the results, and then the average order value (AOV) was calculated. Using SPSS software to perform a non-parametric Friedman test under α = 0.05, the chi-square value of 20.805 was calculated, and looked up the table to get the chi-square critical value of 12.592. Therefore, the hypothesis that “all algorithms have the same performance” was rejected, as the performances of the six algorithms were significantly different.

We used Nemenyi to further test the performance of the algorithm. First, we used Formula (18) to calculate the critical difference (*CD*) of the AOV:(18)$$CD=q_{\alpha }\sqrt{\frac{k(k+1)}{6N}}$$
where *k* is the number of algorithms, *N* is the number of datasets, and *q_α_* can be obtained by looking up the table, where *α* = 0.05. If the difference between the average sequence values of the two algorithms exceeds the *CD*, the assumption that the performance of the two algorithms is the same is rejected. When *k* = 6, *N* = 5, and *q_α_* = 2.850, then *CD* = 3.372.

Figure 5 was drawn according to the ordinal results in Table 3. In Figure 5, the vertical axis shows each algorithm, and the horizontal axis is the average ordinal value. For each algorithm, a dot displays its AOV, and the horizontal line segment with the dot in the center represents the size of the *CD*. If the horizontal line segments of the two algorithms overlap, it means that there was no significant difference between the two algorithms; otherwise, it means that there was a significant difference. It can be easily seen from Figure 5 that there was no significant difference between the algorithms DCRA_M and LSTM-CapsAtt, because their horizontal line segments have overlapping areas. However, the DCRA_M is closer to the vertical axis, so the performance of DCRA_M was better than that of the LSTM-CapsAtt. Similarly, there was no significant difference between the algorithm DNNSN and the LSTM and GELM algorithms. However, the DNNSN is closer to the vertical axis, so the performance of DNNSN was better than LSTM and the GELM. The algorithms DCRA_M and LSTM-CapsAtt do not overlap with the DNNSN, LSTM, and GELM, indicating that the DCRA_M and LSTM-CapsAtt are significantly better than the DNNSN, LSTM, and GELM. DCRA_E is in the middle of all the algorithms, so the performance of DCRA_E was also moderate.

To further verify the fusion effect, the DCRA_E and DCRA_M were compared to the single-modal deep recurrent auto-encoder model. Single-modal deep recurrent auto-encoder input only uses EEG or EOG, and there is no coupling layer. Other hierarchical structures are consistent with the coupling auto-encoder structure. The values of the RMSEs and PCCs of the four methods are listed in Table 4.

As shown in Table 4, the RMSE and PCC values when 0.085 and 0.854, respectively, when EEG was used as the only input to the DRA. When EOG was used as the only input to the deep recurrent auto-encoder (DRA), the RMSE and PCC values were 0.095 and 0.805, respectively. Both the DCRA_E and DCRA_M were better than the single-modal methods, and the improvement is obvious. Because of the intrinsic relationship between EEG and EOG, the DCRA can reinforce the common features of different modes and provide complementary information to achieve better results than a single mode.

At the same time, it can be seen from Table 2 and Table 4 that the fusion methods, including CCRF, CCNF, GELM, DNNSN, and LSTM-CapsAtt, were better than the single-modal method in the test results. It can be said that multi-modal fusion can improve the accuracy of the model and generally performs better than the single-modal method. Overall, the DCRA_M performed better than the other fusion methods.

### 4.2. Analysis of α

In Formula (5), α is the coupling factor of the joint loss function. In order to verify the influence of coupling loss on the joint loss function and RMSE, α values were set as 0, 0.2, 0.4, 0.8, and 1 respectively, and the RMSE values corresponding to different α values were obtained in the experiment, as shown in Figure 6. As can be seen from Figure 6, α values achieved good results at 0.2, 0.4, and 0.8. When α was equal to 0.4, the RMSE value was at the minimum. When α was equal to 0, the coupling loss was 0, and only single-modal loss played a role, which was the worst effect. When α was equal to 1, the coupling loss weight was the largest, and the single-modal loss was 0, so the effect was not good. Theoretically, an α value that is too small will overemphasize the “individuality” of the data and ignore the correlation, while an α value that is too large will overemphasize the correlation and ignore the “individuation” of the data. Therefore, the effect is better when the α value is moderate.

## 5. Conclusions

Vigilance estimation based on EEG and EOG multi-modal data fusion is a hot research topic and has high research value and practical prospects. In this paper, a deep coupling recurrent auto-encoder model that combines EEG and EOG is proposed. This model constructs a coupling layer, which links EEG and EOG together. When constructing the coupling loss function of the model, the Mahalanobis distance is learned by measurements to calculate the similarity of two different modal data. In order to ensure the gradient stability of learning long sequences, a multi-layer GRU is used to construct the auto-encoder model. The deep coupling recurrent auto-encoder model integrates data feature extraction and feature fusion. The results of our experiments show that the proposed method is superior to the single-modal method and the latest multi-modal fusion method. Based on the comparisons of experimental results using different methods, we observed that the proposed method can handle the multimodal data fusion and project the high dimensional vectors of data from different types of sensors into a common latent space, which enables effective classification of multi-model data. However, our method also has some problems, such as the need to take out part of the experimental data to learn a Mahalanobis matrix, and this part of the data must be consistent with the data required for deep model training. At the same time, the Mahalanobis matrix used in the loss function affects the speed of model convergence, and the choice of metric learning method also needs further discussion.

Deep learning has achieved promising results with EEG and EOG fusion, but it also faces some challenges. First of all, there is not a sufficient solution to measure the similarity between different modes, and this area needs more in-depth research and discussion. In addition, our next step is to find a suitable framework for multi-modal fusion.

## Figures and Tables

**Figure 1 entropy-23-01316-f001:**
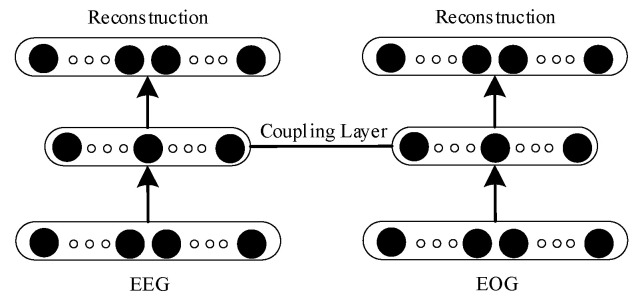
Coupling auto-encoder.

**Figure 2 entropy-23-01316-f002:**
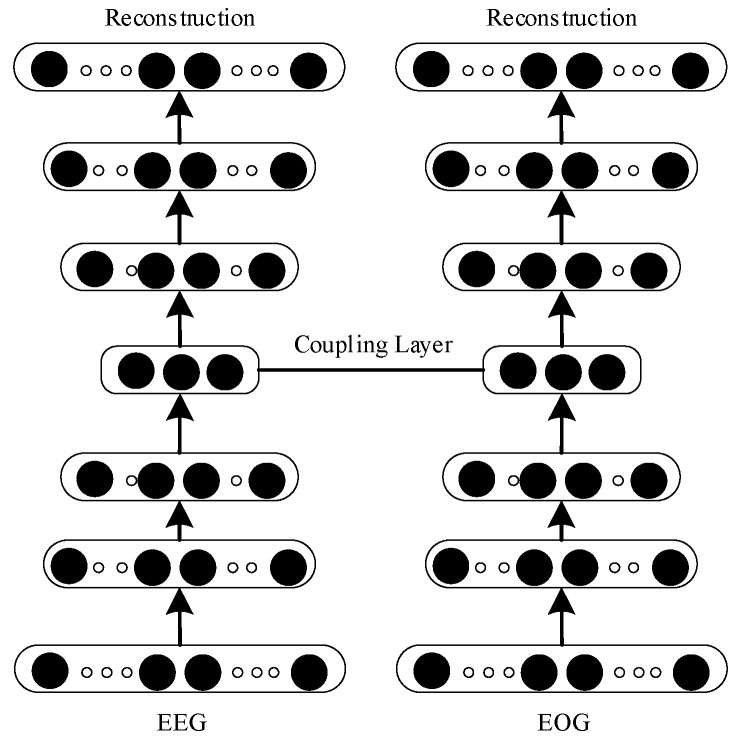
Deep coupling recurrent auto-encoder.

**Figure 3 entropy-23-01316-f003:**
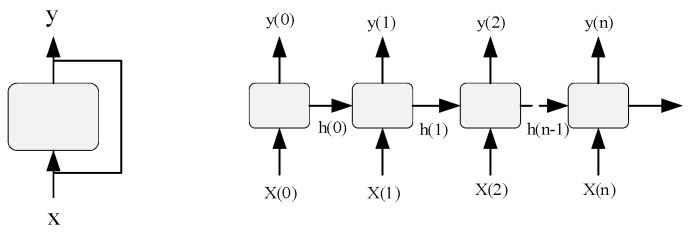
Structure of a recurrent neural network.

**Figure 4 entropy-23-01316-f004:**
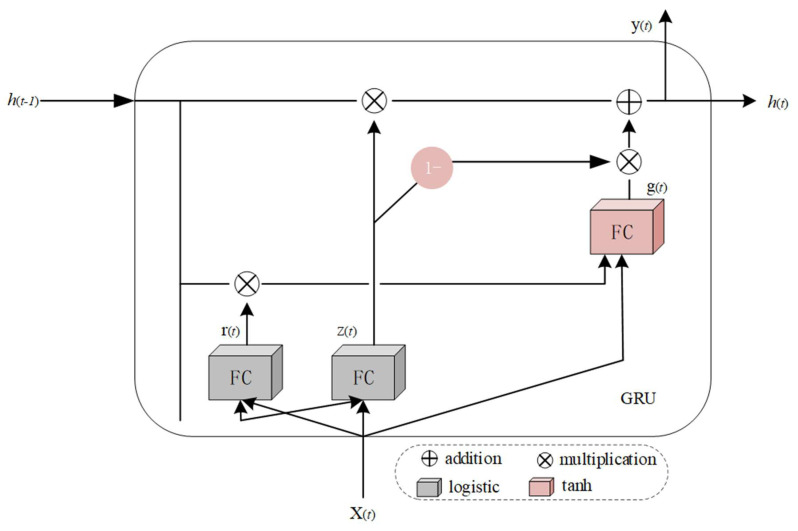
Structure of a GRU.

**Figure 5 entropy-23-01316-f005:**
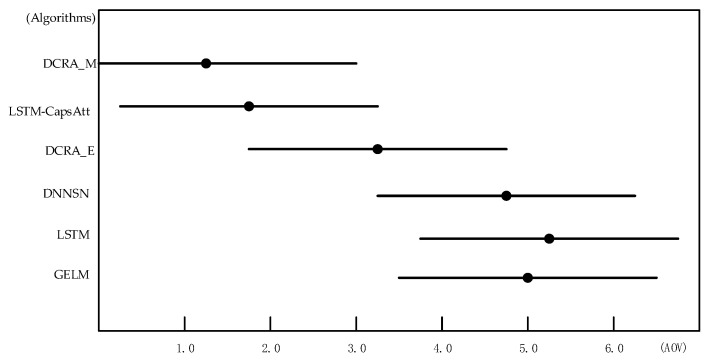
Friedman test chart.

**Figure 6 entropy-23-01316-f006:**
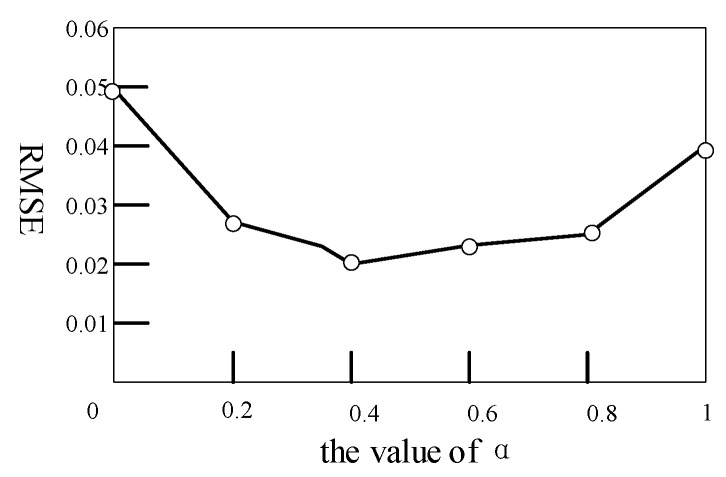
Influence analysis of α value on RMSE.

**Table 1 entropy-23-01316-t001:** Parameter settings of the DCRA model.

Layers	Name	Units (EEG/EOG)	Activation
1	GRU	36/25	Relu
2	GRU	20/20	Relu
3	GRU	16/16	Relu
4	Coupling layer	10	
5	GRU	16/16	Relu
6	GRU	20/20	Relu
7	GRU	36/25	Sigmoid

**Table 2 entropy-23-01316-t002:** Comparison results of different methods.

NO.	Methods	RMSE	PCC
1	SVR	0.100	0.830
2	CCRF	0.100	0.840
3	CCNF	0.095	0.845
4	DAE	0.094	0.852
5	GELM	0.071	0.808
6	LSTM	0.080	0.830
7	DNNSN	0.080	0.860
8	LSTM-CapsAtt	0.029	0.989
9	DCRA_E	0.035	0.980
10	DCRA_M	**0.023**	**0.985**

**Table 3 entropy-23-01316-t003:** The AOVs of 6 algorithms.

Dataset	DRCA_M	LSTM-CapsAtt	DRCA_E	DNNSN	LSTM	GELM
** *D* _1_ **	1	2	3	5	4	6
** *D* _2_ **	1	2	3	4	6	5
** *D* _3_ **	1	2	3	4	5	6
** *D* _4_ **	2	1	4	3	6	5
** *D* _5_ **	1	2	3	6	5	4
**AOV**	1.20	1.80	3.20	4.4	5.20	5.10

**Table 4 entropy-23-01316-t004:** Comparison of effects before and after fusion.

Method	RMSE	PCC
DCRA_M with EEG and EOG	**0.023**	**0.985**
DCRA_E with EEG and EOG	0.035	0.980
DRA with EEG	0.085	0.854
DRA with EOG	0.095	0.805

## Data Availability

In this study, the SEED-VIG dataset was employed, which is freely available in [2].

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
