# Peer review of "Deep Coupling Recurrent Auto-Encoder with Multi-Modal EEG and EOG for Vigilance Estimation"

_entropy, 2021, doi:10.3390/e23101316_

Round 1

Reviewer 1 Report

The original contribution of the paper is not clearly stated in the abstract and introduction. Abstract should include the significance and importance of the work. It should also discuss more about the proposed method. Recent studies from high impact factor journal (see https://www.scimagojr.com/) should be cited like from IEEE transactions, Springer and Elsevier in the introduction or in a related work section. What are the limitations behind this study? This topic should be highlighted somewhere in the text of manuscript. Acronyms and variables in equations must be defined in the article. Many figures are with low resolution manly ih the text part and variables without italic. Please verify it. Table 1 (Parameter settings of DCRA model): Comments about the deep learning models hyperparameters tuning, search space of the variables, computational cost, among others, must be presented. Table 2 (Comparison results of different methods:) A full statistical analysis of the classifiers comparison must be presented based on cross-validation procedure (k-fold), performance measures and significance nonparametric tests. Authors could perform statistical tests (e.g. Friedman test + posthoc Nemenyi test) to compare algorithms and discuss the results in the paper. Conclusion: What are the advantages and disadvantages of this study compared to the existing studies in this area?

Reviewer 2 Report

In the reviewed paper, the authors propose a model of recurrent autoencoder which combines EEG and EOG. The idea seems to be very interesting, but some issues should be improved like:
1) The introduction should be extended. Please indicate the possible application. Moreover, discuss the newest solutions in that matters, for instance, image reconstruction. See such paper-like "Neural image reconstruction using a heuristic validation mechanism", where auto-encoders were used and improved. In addition, add a bullet list of contributions at the end of this section.
2) In Eq (16)-(17), RMSE and PCC were shown. Did you test your proposal on other metrics? Some discussion would be needed.
3) Delete the background from Tab. 1
4) Add some more discussion and analysis of your proposal.
5)Extend the conclusions section to show pros/cons.

Round 2

Reviewer 1 Report

All acronyms must be defined in the article (after your definition in the abstract such as EEG and EOG).

Before of the section 2, the authors could mentioned the organization of the remainder of the article.

L is loss function

->

where L is loss function

More comments about hyperparameters tuning procedure of the models must be presented.

A discussion about the results in Fig. 5 could be presented.

Reviewer 2 Report

Accept in presented form.

Author Response

Thank you very much for your comments. Next, I will modify the format of the article according to the editor's request. Thank you!